# An Alternative Enzymatic Route to the Ergogenic Ketone Body Ester *(R)*-3-Hydroxybutyl *(R)*-3-Hydroxybutyrate

Ferdinando Zaccone [1], Valentina Venturi [1], Pier Paolo Giovannini [1,*], Claudio Trapella [1], Marco Narducci [2], Hugues Fournier [2] and Anna Fantinati [1]

1    Department of Chemistry and Pharmaceutical Sciences, University of Ferrara, Via Luigi Borsari, 46, 44121 Ferrara, Italy; zccfdn@unife.it (F.Z.); vntvnt@unife.it (V.V.); trpcld@unife.it (C.T.); fntnna1@unife.it (A.F.)
2    Impact Science Co., LTD, 3F Toutosui Bldg., 6-22-4 Tsukiji, Chuo-ku, Tokyo 104-0045, Japan; mnardux@gmail.com (M.N.); hugtango@gmail.com (H.F.)
*    Correspondence: pierpaolo.giovannini@unife.it; Tel.: +39-0532-974532

**Abstract:** Recent studies have highlighted the therapeutic and ergogenic potential of the ketone body ester, *(R)*-3-hydroxybutyl-*(R)*-3-hydroxybutyrate. In the present work, the enzymatic synthesis of this biological active compound is reported. The *(R)*-3-hydroxybutyl-*(R)*-3-hydroxybutyrate has been produced through the transesterification of racemic ethyl 3-hydroxybutyrate with *(R)*-1,3-butanediol by exploiting the selectivity of *Candida antarctica* lipase B (CAL-B). The needed *(R)*-1,3-butanediol was in turn obtained from the kinetic resolution of the racemate achieved by acetylation with vinyl acetate, also in this case, thanks to the enantioselectivity of the CAL-B used as catalyst. Finally, the stereochemical inversion of the unreacted *(S)* enantiomers of the ethyl 3-hydroxybutyate and 1,3-butanediol accomplished by known procedure allowed to increase the overall yield of the synthetic pathway by incorporating up to 70% of the starting racemic reagents into the final product.

**Keywords:** ketone body ester; lipase; kinetic resolution; asymmetric synthesis; configuration inversion



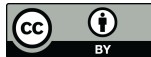

## 1. Introduction

The ketone bodies *(R)*-3-hydroxybutyrate and acetoacetate, are short chain acids produced by the liver from the free fatty acids released from adipose tissue. The blood ketone bodies concentration normally ranges below 1 mM [1] increasing up to 5–7 mM during prolonged fasts [2]. Under this metabolic condition, known as ketosis, ketone bodies efficiently replace glucose as respiratory substrate, furnishing a higher adenosine triphosphate (ATP) yield with respect to pyruvate, the end-product of glycolysis [3]. This explain why a mild ketosis is beneficial for muscle and brain during prolonged physical exercise [4–6]. Furthermore, significant results in the treatment of patients affected by neurodegenerative diseases [1,7–9] and epilepsy [10] have been obtained through the increasing of blood ketone bodies induced by consumption of a ketogenic diet. However, a nutrition devoid of carbohydrate and rich of saturated fats is scarcely tolerated by most of the patients, and increases plasma cholesterol and free fatty acids, both known risk factors for several pathologies [11,12]. On the other hand, administration of therapeutically relevant amounts of the ketone bodies as free acids or sodium salts resulted in dangerous acidosis or sodium overload, respectively [13]. The oral assumption of ketone bodies esters has been demonstrated as a successful alternative to induce beneficial levels of ketosis avoiding the increase of blood levels of cholesterol and fatty acids as well as the risk of acidosis or sodium overloading [14]. The most employed ketone body ester is the palatable and nontoxic *(R)*-3-hydroxybutyl *(R)*-3-hydroxybutyrrate [15,16] which is cleaved in vivo to *(R)*-3-hydroxybutyrate and *(R)*-1,3-butanediol. The former is the most abundant ketone body of the entire circulating pool (about 70%) [4], the latter is converted to acetoacetate

and *(R)*-3-hydroxybutyrate in the liver [17]. The *(R)*-3-hydroxybutyl *(R)*-3-hydroxybutyrate has been produced by enzymatic reduction of 3-oxobutyl acetoacetate (in turn obtained by transesterification of diketene with 4-hydroxybutan-2-one) [18]. The fermentative production by means of metabolically engineered anaerobic microorganisms has been also reported [19]. The simplest strategy for producing this ketone body ester is the lipase-catalyzed transesterification of ethyl *(R)*-3-hydroxybutyrate with *(R)*-1,3-butandiol [20]. This approach requires enantiopure reagents. The *(R)*-3-hydroxybutyrate can been obtained by enzymatic kinetic resolution of the racemate [21] as well as by alcoholysis of polyhydroxybutyrate, a polyester produced on large scale by bacterial fermentation [22]. Recently, *(R)*-3-hydroxyburate and *(R)*-1,3-butandiol have been respectively obtained by acid catalyzed ethanolysis or sodium borohydride reduction of *(R)*-β-butyrolactone deriving from enzymatic hydrolysis of the corresponding racemate (Scheme 1) [23,24]. Herein we report a new enzymatic approach which, starting from both racemic ethyl 3-hydroxybutyrate and 1,3-butandiol, affords the ketone body ester *(R)*-3-hydroxybuthyl *(R)*-3-hydroxybutyrate. The overall yield of the synthetic pathway was pushed up to 70% thanks to the recycling of the *(S)* reagents by stereochemical inversion.

**Scheme 1.** Synthesis of *(R)*-hydroxybuthyl *(R)*-3-hydroxybutyrate **4** starting from racemic β-butyrolactone **1** following the methodology reported in reference 24. Reaction conditions: Step (**a**) racemic-**1** (50 mmol), $H_2O$ (30 mmol), methyl tert-butyl ether (MTBE) (250 mL), *Candida antarctica* lipase B (CAL-B) (0.3 g), 25 °C, 2 h; Step (**b**) *(R)*-**1** (23 mmol), ethanol (50 mL), $H_2SO_4$ (0.2% *v/v*), 25 °C, 48 h; Step (**c**) *(R)*-**2** (20 mmol), *(R)*-**3** (20 mmol), CAL-B (0.2 g), 30 °C, 80 mm Hg, 6 h. Yields of *(R)*-**2** and *(R,R)*-**4** referred to the isolated products; the yield of compound *(S)*-**2** was deduced from the CG analysis of the crude reaction mixture.

## 2. Results and Discussion

### 2.1. Synthesis of (R)-3-Hydroxybutyl (R)-3-Hydroxybutyrate from Enantioenriched (R)-3-Hydroxybutyrate

To produce *(R)*-3-hydroxybutyl *(R)*-3-hydroxybutyrate on a gram scale, we followed the procedure recently proposed by Ulrich and coworkers [24]. In this procedure, the racemic β-butyrolactone (compound **1**, Scheme 1) was kinetically resolved by *Candida antarctica* lipase B (CAL-B) catalyzed hydrolysis. After aqueous workup to remove the *(S)*-3-hydroxybutanoic acid, the resulting *(R)*-β-butyrolactone was transesterified with ethanol to give ethyl *(R)*-3-hydroxybutyrate **2** (steps a) and b), Scheme 1). In our results, these reaction sequences gave *(R)*-**2** with 85% enantiomeric excess (ee). The lower optical purity with respect to the literature data (>99%) [24], was probably due to an incomplete hydrolytic step. Despite this, we submitted the enantioenriched *(R)*-**2** to the CAL-B-catalyzed transesterification with *(R)*-1,2-butandiol **3** (step c), Scheme 1). Following the reaction course by chiral phase gas chromatographic analysis, we noted that, once the complete conversion of *(R)*-**2** to the desired *(R)*-3-hydroxybuthyl *(R)*-3-hydroxybutyrate *(R,R)*-**4** was reached, the small amount of *(S)*-**2** (7.5%) present in the starting ethyl ester remained unreacted. This prompted us to investigate the possibility to directly use racemic-**2** for the enantioselective synthesis of the ketone body ester *(R,R)*-**4**.

### 2.2. Synthesis of (R)-3-Hydroxybutyl (R)-3-Hydroxybutyrate from Racemic 3-Hydroxybutyrate

The possibility to produce the ketone body ester *(R,R)*-**4** starting from the racemic ethyl ester **2** was verified by reacting *(R)*-1,3-butandiol (**3**) with two equivalents of racemic-**2** in

the presence of CAL-B without the addition of any solvent. The reaction was gently shaken at 30 °C under reduced pressure (80 mmHg) in order to remove of the coproduced ethanol. The formation of product **4**, as well as its stereochemistry, were periodically checked by chiral phase GC analysis. After 5 h, the diol **3** was completely converted to the expected (*R,R*)-**4** leaving the ethyl ester (*S*)-**2** unreacted (Scheme 2, route a). After removing the enzyme by filtration, the crude reaction mixture was distilled under vacuum to recover (*S*)-**2** (40% yield, >91% ee) as the distillate and (*R,R*)-**4** (48% yield, >20/1 dr) as the residue.

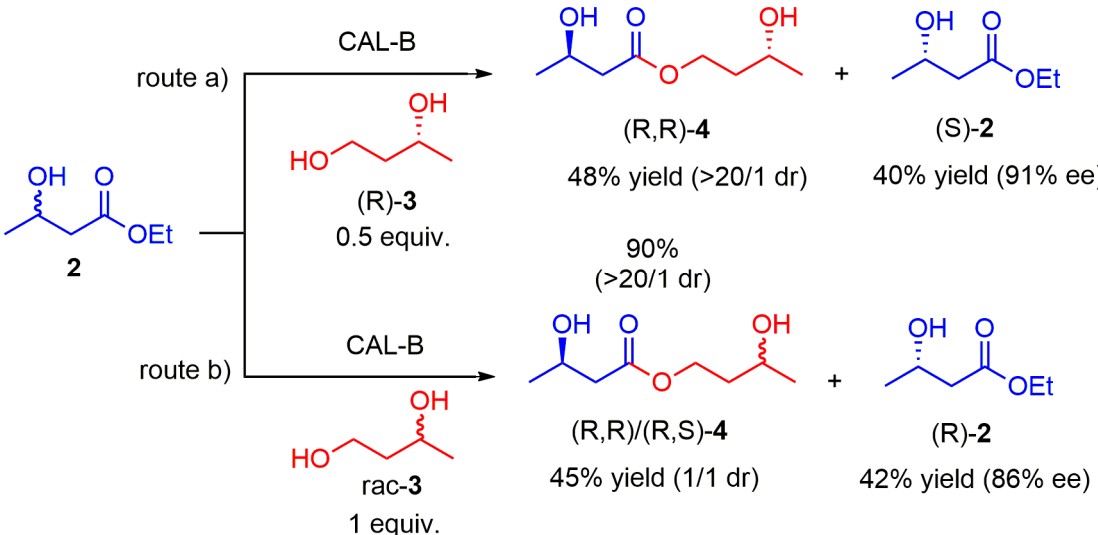

**Scheme 2.** CAL-B catalyzed transesterification of racemic ethyl 3-hydroxybutyrate **2** with (*R*)- or racemic 1,3-butandiol **3** (route (**a**) and (**b**), respectively). Reaction conditions: (*R*)-**2** (20 mmol), (*R*)-**3** (20 mmol, route a) or racemic-**3** (40 mmol, route b), CAL-B (0.2 g), 30 °C, 80 mm Hg, 6–8 h. Yields for routea (a) referred to the isolated products. Yields for route (b) have been deduced by gas chromatographic (GC) analysis.

The transesterification reaction between both the racemic **2** and **3**, was attempted as well (Scheme 3, route b). However, in this case, because of the distance between the reactive hydroxyl group and the chiral carbon (C3) both the enantiomers of the diol **3** reacted with comparable rates. As a result, a 1:1 mixture of (*R,R*)- and (*R,S*)-**4** was obtained (see Supplementary Materials).

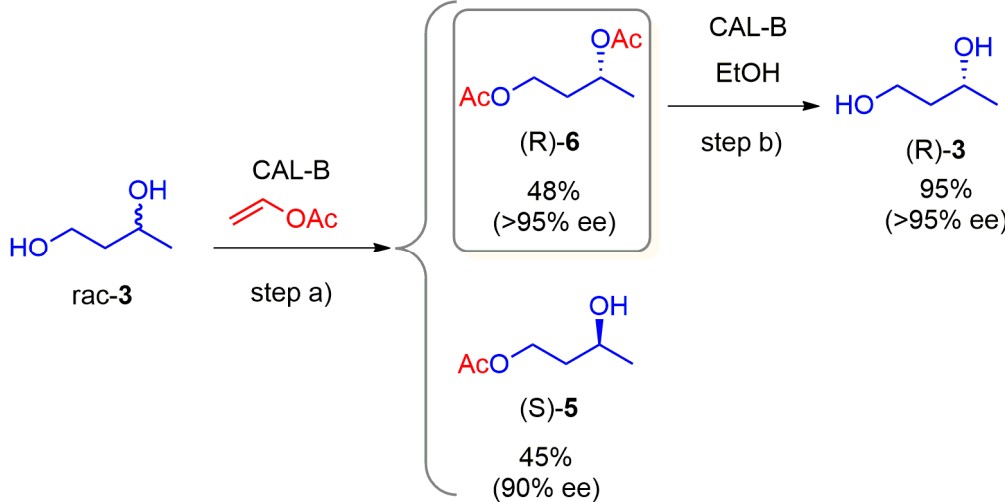

**Scheme 3.** The synthetic pathway for the preparation of enantiopure (*R*)-1,3-butanediol **3**. Reaction condition: step (**a**) racemic-**3** (20 mmol), vinyl acetate (30 mmol), CAL-B (0.2 g), 30 °C, 2.5 h; step (**b**) (*R*)-**6** (9.6 mmol), ethanol (28.8 mmol), CAL-B (0.2 g), cyclohexane (10 mL), 30 °C, 2h. Yields referred to isolated products.

### 2.3. CAL-B Catalyzed Kinetic Resolution of the 1,3-Butandiol

Once we ascertained the possibility to use ester **2** as racemate, as well as the needing of enantiomerically pure *(R)*-**3**, we engaged the study on the kinetic resolution of the racemic diol **3**.

The structural resemblance of diol **3** and ester **2**, suggested us to attempt the kinetic resolution of the former, through an enzymatic approach developed for the later [21]. On the other hand, a precedent study reported the lipase mediated kinetic resolution of racemic-**3** with Chirazyme[TM][25]. The reaction, once again catalyzed by CAL-B, was performed in a solvent-free system with 1.5 equivalents of vinyl acetate as the acylating agent. The time course of the reaction monitored by chiral phase CG analysis (Figure 1) showed the not stereoselective esterification of the primary alcoholic group leading to the complete conversion of the racemic diol **3** to *(R)-* and *(S)*-3-hydroxybutyl acetate **5** within the first half-hour. After this, the concentration of *(S)*-**5** remained almost unvaried, while *(R)*-**5** was quickly converted to *(R)*-1,3-butandiol diacetate **6**. The reaction performed on preparative scale (1 g of racemic-**3**) gave after 2.5 h the complete conversion of the racemic diol **3** to an almost equimolar mixture of *(S)*-**5** and *(R)*-**6** (Scheme 3). After removing the vinyl acetate by evaporation, the crude reaction mixture was chromatographed on silica gel to separate *(S)*-**5** (45% yield, 90% ee) from *(R)*-**6** (48% yield, >95% ee).

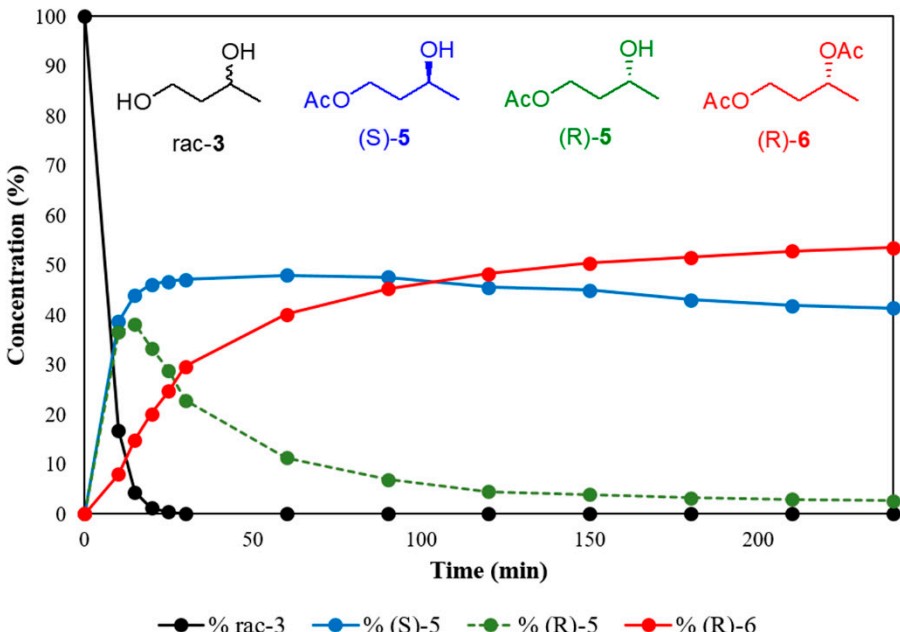

**Figure 1.** Time course of the CAL-B catalyzed kinetic resolution of racemic diol **3**.

The diacetate was then subjected to ethanholysis in the presence of CAL-B. The reaction was conducted in cyclohexane as the solvent since the use of pure ethanol was reported as detrimental for the stability of the enzyme [21]. After evaporation of cyclohexane, excess of ethanol and ethyl acetate coproduct, the *(R)*-**3** was obtained in 95% yield (>95 % ee) and used without further purification for the synthesis of (R,R)-**4**.

### 2.4. Inversion of Configuration of (S)-3-Hydroxybutyl Acetate 5

The overall yield of the synthetic pathway, including the preparation of the enantiopure diol *(R)*-**3**, was 38% (calculated on the starting racemic-**3**). Therefore, in order to increase the overall yield as well as the economy of the process, the configuration inversion of the coproducts ethyl *(S)*-hydroxybutyrate **2** and *(S)*-3-hydroxybutyl acetate **5** was then taken into account. The inversion of *(S)-* to *(R)*-**2** by mesylation of the hydroxyl group followed by $S_N2$ with cesium acetate has been recently published [26]. However, a following work reported a low selectivity of this procedure because of the formation of ethyl

3-methylacrylate as elimination by-product (Scheme 4) [24]. For this reason, we focused alternative inversion strategy, based on the tosylation of the hydroxyl group followed by S$_N$2 inversion with triethylammonium acetate (Scheme 4) [27]. This approach provided the expected O-acetylated *(R)*-**2** in 70% yield (88% ee), a result in line with that reported in the original study. The acetylated product was finally converted to *(R)*-**2** by ethanolysis in the presence of CAL-B as described [21].

**Scheme 4.** Synthetic pathways for the configuration inversion of *(S)*-**2** through mesylation followed by S$_N$2 with cesium acetate [24] or by means of tosylation followed by S$_N$2 with triethylammonium acetate [27].

Once verified the efficiency of the method as well as its compatibility with the ester group, the *(S)*-3-hydroxybutyl acetate **5** was submitted to the same procedure. The reaction of *(S)*-**5** with *p*-toluenesulfonyl chloride in pyridine gave the expected tosyl derivative **9** in 96% yield (Scheme 5). The following S$_N$2 was performed by adding compound **9** to a solution of triethylamine and acetic acid in toluene and warming the resulting mixture for 4h at 80 °C. After aqueous workup and solvent evaporation, the residue was chromatographed to give *(R)*-**6** in 80% yield (88% ee).

**Scheme 5.** Inversion of the configuration of the *(S)*-3-hydroxybutyl acetate **5**. Reaction condition: step (**a**) *(S)*-**5** (7.5 mmol), *p*-dimethylaminopyridine (0.38 mmol), pyridine (5 mL), *p*-toluenesulfonyl chloride (9.6 mmol, added at 0° C), 25 °C, 5 h; step (**b**) *(S)*-**9** (7.2 mmol), triethylamine (2.16 mmol), AcOH (13.0 mmol), toluene (5 mL), 80 °C, 4 h. The yields referred to crude product **9** and isolated product **6**. The ee of compound **9** has not determined (n.d.).

## 3. Materials and Methods

### 3.1. General Information

All commercially available reagents were used as received without further purification, unless otherwise stated. The CAL-B Novozym®435 was purchased from Novozymes (>Copenaghen, Denmark). Reactions were monitored by TLC on silica gel 60 F254 with detection by charring with phosphomolybdic acid. Flash column chromatography was performed on silica gel 60 (230–400 mesh). $^1$H and $^{13}$C nuclear magnetic resonance (NMR) spectra were recorded on 300 and 400 MHz Varianspectrometers (Palo Alto, CA, USA) at room temperature using CDCl$_3$ as solvent. Chemical shifts (δ) are reported in ppm relative to residual solvent signals. Optical rotations were measured at 20 ± 2 °C in the stated solvent; [α]$^{20}_D$ values are given in $10^{-1}$ deg cm$^2$ g$^{-1}$. High-resolution mass spectra

(HRMS) were recorded in positive ion mode with an Agilent 6520 high performance liquid chromatography (HPLC)-Chip coupled with a quadrupole/time of fly-mass spectrometer (Q/TOF-MS) nanospray system unit (Santa Clara, CA, USA) to produce spectra. GC analyses were performed using a Thermo Focus gas chromatograph (Waltham, MA, USA) equipped with a flame ionization detector and a Megadex 5 column (25 m × 0.25 mm), with the temperature programs as specified.

### 3.2. Gas Chromatographic Analysis

Samples (5 mg) were diluted with ethyl acetate and injected (1 μL). The products were detected using the following temperature program: 70 °C for 15 min, 10° C/min up to 200 °C. $R_T$ for ester **2**: 18.5 min; $R_T$ for diol **3**: 20.7 min; $R_T$ for ketone body ester *(S,R)*-**4**: 22.0 min; $R_T$ for ketone body ester *(R,R)*-**4**: 22.0 min. For a better separation of the enantiomers, ester **2** and diol **3** were converted to the corresponding *O*-acetyl derivatives before the injection. The sample (5 mg) was diluted with acetic anhydride (20 μL) and triethylamine (5 μL) and kept at room temperature for two hours. The mixture was diluted with ethyl acetate (1 mL) and injected (1 μL) using the following temperature program: from 60 °C 2 °C/min up to 200 ° C. $R_T$ for the acetyl derivative of *(S)*-**2**: 18.7 min; $R_T$ for the acetyl derivative of (2)-**2**: 21.3 min. $R_T$ for diacetate *(S)*-**6**: 23.1 min; $R_T$ for diacetate *(R)*-**6**: 25.1 min.

### 3.3. Synthesis of (R)-3-Hydroxybutyl (R)-3-Hydroxybutyrate 4 From Racemic 3-Hydroxybutyrate

A mixture of racemic ethyl ester **2** (1 g, 7.6 mmol), *(R)*-1,3 butandiol **3** (0.34 g, 3.9 mmol) and CAL-B (70 mg) was gently shaken under reduced pressure (80 mmHg) at 30 °C for 6 h. The reaction mixture was filtered to remove the enzyme and evaporated under reduced pressure (80 mm Hg) to separate unreacted ethyl ester *(S)*-**2** as the distillate (0.4 g, 3.0 mmol), 40% yield (91% ee), from the *(R)*-3-hydroxybutyl *(R)*-3-hydroxybutyrate *(R,R)*-**4** (0.64 g, 3.6 mmol), 48% yield (>90% dr). [1]H NMR (300 MHz, CDCl$_3$) δ 4.34–4.25 (m, 1H, CHOH), 4.22–4.10 (m, 2H, CH$_2$OCO), 3.93–3.79 (m, 1H, CHOH), 2.45 (dd, *J* = 16.1, 3.9 Hz, 1H, CH$_2$CO$_2$), 2.39 (dd, *J* = 16.1, 8.4 Hz, 1H, CH$_2$CO$_2$), 1.82–1.63 (m, 2H, CH$_2$), 1.19 (d, *J* = 2.7 Hz, 3H, CH$_3$), 1.18 (d, *J* = 2.6 Hz, 3H, CH$_3$). [13]C NMR (100 MHz, CDCl$_3$) δ 172.9, 65.1, 54.6, 62.1, 43.1, 37.6, 23.5, 22.6. HRMS (ESI) m/z calcd for C$_8$H$_{17}$O$_4^+$: 177,1127 [M + H]$^+$; found: 177,1137.

### 3.4. Kinetic Resolution of Racemic-1,3-Butandiol 3

A mixture of racemic 1,3-butandiol **3** (1.8 g, 20 mmol), vinyl acetate (2.6 g, 30 mmol) and CAL-B (0.2 g) was gently shaken at 30 °C following the reaction course by chiral phase GC analysis. The reaction was stopped when the diol **3** was completely converted (about 2.5 h). The mixture was diluted with methylene chloride (10 mL) and filtered to remove the enzyme. After evaporation of the solvent the residue was chromatographed on silica gel with cyclohexane-ethyl acetate-methanol (15:4:1) as the eluent. *(S)*-3-hydroxybutyl acetate **5** (1.19 g, 9.0 mmol), 45% yield, (90% ee); [α]$_D^{20}$ = + 19.1 (c 2.0, CHCl$_3$), lit + 17.5 (c 1.4) [25]. [1]H NMR (300 MHz, CDCl$_3$) δ 4.38–4.27 (m, 1H, CHOAc), 4.16 – 4.07 (m, 1H, CHOAc), 3.95–3.88 (m, 1H, CHOH), 2.05 (s, 3H, Ac), 1.85–1.62 (m, 2H, CH$_2$), 1.22 (d, *J* = 6.2 Hz, 3H, CH$_3$). [13]C NMR (100 MHz, CDCl$_3$) δ 171.4, 64.6, 61.7, 37.8, 23.4, 20.9. HRMS (ESI) m/z calcd for C$_6$H$_{13}$O$_3^+$: 133,0865 [M + H]$^+$; found: 133,0858. *(R)*-1,3-butanediol diacetate **6** (1.67 g, 9.6 mmol), 48 % yield, (>95% ee); [α]$_D^{20}$ = −25.7 (c 2.0, CHCl$_3$), lit + 23.5 (c 1.4) [25]. [1]H NMR (300 MHz, CDCl$_3$) δ 4.96–4.83 (m, 1H, CHOAc), 4.05–3.95 (m, 2H, CH$_2$OAc), 1.93 (s, 3H, Ac), 1.92 (s, 3H, Ac), 1.86–1.67 (m, 2H, CH$_2$), 1.14 (d, *J* = 6.2 Hz, 3H, CH$_3$). [13]C NMR (100 MHz, CDCl$_3$) δ 170.8, 170.3, 67.7, 60.6, 34.6, 21.1, 20.7, 19.9. HRMS (ESI) m/z calcd for C$_8$H$_{15}$O$_4^+$: 175,0970 [M + H]$^+$; found: 175,0981.

The *(R)*-1,3-butanediol diacetate **6** (1.67 g, 9.6 mmol) was dissolved in cyclohexane (10 mL). Ethanol (1.32 g, 28.8 mmol) and CAL-B (0.2 g) were added and the mixture was gently shaken at 30 °C following the reaction course by TLC. When the diacetate **6** was fully

converted to the diol **3** the reaction was filtered and evaporated to afford *(R)*-1,3-butanediol **3** (0.82 g, 9.1 mmol), 95% yield, (>95% ee).

*3.5. Inversion of Configuration of (S)-3-Hydroxybutyl Acetate 5*

A solution of (*S*)-3-hydroxybutyl acetate **5** (1 g, 7.5 mmol) and *p*-dimethylaminopyridine (47 mg, 0.38 mmol) in pyridine (5 mL) was cooled to 0 °C and *p*-toluenesulfonyl chloride (1.8 g, 9.6 mmol) was added in portions over 30 min. The mixture was kept at room temperature for 5 h and then diluted with water (16 mL). The white solid precipitated was filtered, washed with cold water (2 × 10 mL) and dried under vacuum at 40 °C to give compound **9** (2.06 g, 7.2 mmol), 96% yield; [1]H NMR (300 MHz, CDCl$_3$) δ 7.79 (d, J = 8.3 Hz, 2H, Ar), 7.32 (d, J = 8.4 Hz, 2H, Ar), 4.80–4.67 (m, 1H, CHOTs), 4.07–3.97 (m, 1H, CHOAc), 3.95–3.85 (m, 1H, CHOAc), 2.44 (s, 3H, Ts), 1.96 (s, 3H, Ac), 1.99–1.80 (m, 2H, CH$_2$), 1.34 (d, J = 6.2 Hz, 3H, CH$_3$). [13]C NMR (100 MHz, CDCl$_3$) δ 171.0, 145.0, 134.5, 130.1, 128.1, 77.1, 60.4, 35.8, 21.9, 21.4, 21.1. The crude compound **9** (2.06 g, 7.2 mmol) was added to a solution of triethylamine (0.22 g, 2.16 mmol) and acetic acid (0.78 g, 13 mmol) in toluene (5 mL) previously stirred at room temperature for half an hour. The mixture was heated to 80 °C, and stirred at this temperature for 4 h. After cooling to room temperature, the reaction mixture was diluted with toluene (40 mL) and was washed successively with aqueous 2 M HCl solution (20 mL) and 10% (w/v) aqueous K$_2$CO$_3$ solution (30 mL). The organic layer was separated, dried over anhydrous Na$_2$SO$_4$ and evaporated to afford the 1,3-butanediol diacetate **6** (1.0 g, 5.76 mmol), 80% yield, (88% ee).

## 4. Conclusions

This enzymatic methodology allows for easy access to the nutraceutical and pharmaceutical relevant *(R)*-3-hydroxybutyl *(R)*-3-hydroxybutyrate starting from cheap, racemic reagents. The ethyl 3-hydroxybutyrate was used directly in racemic form while the needed *(R)*-1,3-butandiol was obtained by enzymatic kinetic resolution of the corresponding racemate. Thanks to the configuration inversion of both the distomers *(S)*-3-hydroxybutyrate and *(S)*-1,3-butandiol, the overall yield of the process has been increased over the classical 50% normally achieved by kinetic resolution-based methodologies.

**Supplementary Materials:** The following are available online at https://www.mdpi.com/2073-4344/11/1/140/s1, Figure S1: [1]H- and [13]C-NMR spectra of compound **4**; Figure S2: [1]H- and [13]C-NMR spectra of compound **5**; Figure S3: [1]H- and [13]C-NMR spectra of compound **6**; Figure S4: [1]H- and [13]C-NMR spectra of compound **9**; Figure S5: Chiral phase GC of (*R,R*)-**4** and (*R,R*)/(*R,S*)-**4** mixture; Figure S6: Chiral phase GC of acetylated (*R*)-**2** from (*S*)-**2** inversion; Figure S7: Chiral phase GC of acetylated *(R)*-**3** from kinetic resolution rac-**3**; Figure S8: Chiral phase GC of *(R)*-**6** from inversion of *(S)*-**5**.

**Author Contributions:** Conceptualization, P.P.G. and A.F.; methodology, F.Z. and H.F.; investigation, V.V. and C.T.; resources, M.N.; funding acquisition, M.N. All authors have read and agreed to the published version of the manuscript.

**Funding:** This research was funded by University of Ferrara, grant FAR 2020.

**Institutional Review Board Statement:** Not applicable.

**Informed Consent Statement:** Not applicable.

**Data Availability Statement:** Data are contained within the article.

**Acknowledgments:** We gratefully thank Paolo Formaglio for NMR experiments and Francesco Presini for GC analyses.

**Conflicts of Interest:** The authors declare no conflict of interest.

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
