# Peer review of "An Alternative Enzymatic Route to the Ergogenic Ketone Body Ester (R)-3-Hydroxybutyl (R)-3-Hydroxybutyrate"

_catalysts, doi:10.3390/catal11010140_

Round 1
Reviewer 1 Report
The manuscript submitted by Giovannini et al describes an enzymatic route to the (R)-3-Hydroxybutyl (R)-3-Hydroxybutyrate, which is a very important blood ketone body. The synthetic route is based on the use of several enzymatic procedures to achieve the desired enantio-rich products from racemic starting materials. Moreover, the epimerization of undesired enantiomers allows to increase the global yield of the present procedure. The alternative route showed in this manuscript, under my point of view, is very important because allows obtained the final ester in high enantiomerical excess and yield using easy synthetic procedures.
The manuscript is well written and is quite clear, although would be necessary to include a scheme after line 170 to make the aforementioned more visible.
In addition, some mistakes should be corrected:
- The absolute configuration, R and S, must be write in italic format.
- Line 22: Change “1,3-buatnediol” by “1,3-butanediol”
- Line 50: Change “(R)-3-hydroxybutyrrate” by “(R)-3-hydroxybutyrate”
- Line 151: Change “(R)-1,3-buatndiol 3” by “(R)-1,3-butanediol 3”
- Line 204: Change “Gaschromatographic analysis” by “Gas chromatographic analysis”
- Line 223: Change “48% yiled” by “48% yield”
- Line 247: Change “where” by “were”
- Line 253: Change “A solution of (S)-3-hyroxybutyl diacetate 6” by “A solution of (S)-3-hydroxybutyl acetate 5”
- Line 261: Change “13C NMR (100 MHz, CDCl3) δ 180.0, 145.0,..by “13C NMR (100 MHz, CDCl3) d 171.0, 145.0,..”
Regarding the experimental section, in the 1H NMR spectra, the chemical shift of a multiplet signal is a range, not a simple value. Please, corrected it.
Taking into account the comments above, I consider that the manuscript could be published in Catalysts after make the changes suggested.

Author Response
RESPONSE TO REFEREE 1
Referee 1: - The manuscript is well written and is quite clear, although would be necessary to include a scheme after line 170 to make the aforementioned more visible.
Response - A new Scheme 4, together with the corresponding footnote, has been added after line 170 of the old manuscript (line 167 of the revised one) as required by the Referee 1 in order to clarify what reported in the above text about the two known approaches for the configuration inversion of (S)-2. As a consequence, the old Scheme 4 becomes Scheme 5 in the revised manuscript. The term “Scheme 4”, in brackets has been added in the text at lines 161 and 163 of the revised manuscript. The term “Scheme 4” has been changed with “Scheme 5” at line 176 of the revised manuscript (line 174 of the old one).
Referee 1 - The absolute configuration, R and S, must be write in italic format.
Response - The absolute configuration, R and S, have placed in italic format in the text (highlighted), in schemes and figures and in the Supplementary material as well (not highlighted).
Referee 1 - Line 22: Change “1,3-buatnediol” by “1,3-butanediol”
Response - The term “1,3-buatnediol” has been corrected to “1,3-butanediol” in the abstract (line 19 of the revised manuscript; line 22 of the old one). We inform Referee 1 that the abstract has been fully revised according to suggestion of Referee 2.
Referee 1 - Line 50: Change “(R)-3-hydroxybutyrrate” by “(R)-3-hydroxybutyrate”
Response - The term “(R)-3-hydroxybutyrrate” has been changed to “(R)-3-hydroxybutyrate” at line 46 of the revised manuscript (line 50 of the old one).
Referee 1 - Line 151: Change “(R)-1,3-buatndiol 3” by “(R)-1,3-butanediol 3”
Response - The term “(R)-1,3-buatndiol 3” has been changed to “(R)-1,3-butanediol 3” in the footnote of Scheme 3 (line 147 of the revised manuscript; line 151of the old one).
Referee 1 - Line 204: Change “Gaschromatographic analysis” by “Gas chromatographic analysis”
Response - The title of section 3.2 has been corrected: the term “Gaschromatographic analysis” has been replaced with “Gas chromatographic analysis” (line 206 of the revised manuscript - line 204 of the old one).
Referee 1 - Line 223: Change “48% yiled” by “48% yield”
Response - At line 225 of the revised manuscript (line 223 of the old one) “48% yiled” has been changed with “48% yield”
Referee 1 - Line 247: Change “where” by “were”
Response - At line 250 of the revised manuscript (line 247 of the old one) “where” has been replaced with “were”.
Referee 1 - Line 253: Change “A solution of (S)-3-hyroxybutyl diacetate 6” by “A solution of (S)-3-hydroxybutyl acetate 5”
Response - At the beginning of section 3.2 (line 256 of the revised manuscript; line 253 of the old one) “A solution of (S)-3-hyroxybutyl diacetate 6” has been corrected to “A solution of (S)-3-hydroxybutyl acetate 5”.
Referee 1 - Line 261: Change “13C NMR (100 MHz, CDCl3) δ 180.0, 145.0,..by “13C NMR (100 MHz, CDCl3) d 171.0, 145.0,..”
Response - The 13C NMR of compound 9 has been checked and modified by changing δ 180.0 with δ 171.0 (line 264 of the revised manuscript; line 261 of the old version). The corresponding spectrum reported in the supporting information has been slightly modified as well by removing from the peck list some weak signals not attributable to the compound.
Referee 1 - Regarding the experimental section, in the 1H NMR spectra, the chemical shift of a multiplet signal is a range, not a simple value. Please, corrected it.
Response - The 1H NMR spectra of compounds 4 (line 225-228 new version), 5 (line 240-241 new version), 6 (line 245-246 new version), 9 (line 261-264 new version) have been checked and corrected according to the notes of Referees 1 and 2.
Reviewer 2 Report
This is a short paper devoted to the synthesis of (R)-3-hydroxybutyl (R)-3-hydroxybutyrate through the enantioselective transesterification of racemic ethyl 3-hydroxybutyrate with (R)-1,3-butanediol catalyzed by immobilized Candida antarctica lipase B (CAL-B).
The catalytic studies are well done, and the results are supported by the experimental data. I recommend the manuscript for publication after revision.
The authors should consider the following:
The abstract should be rewritten. The first three sentences are more suitable for an introduction than an abstract. Also, the stereochemical inversion achieved by tosylation and SN2 reaction with ammonium acetate is a well-known procedure. Therefore, this information must either be removed from the abstract or shortened.
The assignment of 1H NMR spectra should be carefully checked. Also, the ranges of chemical shifts should be specified for multiplet signals.
Several formulas in the Supporting information are distorted.
Author Response
RESPONSE TO REFEREE 2
Referee 2 - The abstract should be rewritten. The first three sentences are more suitable for an introduction than an abstract. Also, the stereochemical inversion achieved by tosylation and SN2 reaction with ammonium acetate is a well-known procedure. Therefore, this information must either be removed from the abstract or shortened.
Response - The abstract has been rewritten according to the suggestion of Referee 2 by condensing the importance of the target product into a single short phrase at the beginning of the abstract and by giving less emphasis to the known methodology adopted for the configuration inversion of the unreacted enantiomers of 1,3-butanediol and ethyl 3-hydroxybutyrate.
Referee 2 - The assignment of 1H NMR spectra should be carefully checked. Also, the ranges of chemical shifts should be specified for multiplet signals.
Response - The 1H NMR spectra of compounds 4 (line 225-228 new version), 5 (line 240-241 new version), 6 (line 245-246 new version), 9 (line 261-264 new version) have been checked and corrected according to the notes of Referees 1 and 2 notes.
Referee 2 - Several formulas in the Supporting information are distorted.
Response - New formulas have been pasted in the NMR spectra of the Supporting information.
Round 2
Reviewer 2 Report
The following minor corrections are necessary.
1) Scheme 4. Please mark the O atoms at the up wedges of the O-acetylated-(R)-2 molecules in red color.
2) Scheme 5. Please mark the O atom at the up wedge of (R)-6 in red color.
3) Rows 245-246. Although the signals of the two Ac groups are overlapped, it is possible to distinguish the maxima of the singlet signals in the 1H NMR spectrum.
I recommend the manuscript for publication in Catalysts after these minor corrections.
Author Response
We grateful thanks the Reviewer for the accurated notes. The following corrections have been made:
Reviewer - Scheme 4. Please mark the O atoms at the up wedges of the O-acetylated-(R)-2 molecules in red color.
Authors response - The cheme 4 has been modified according to the accurate note of the Reviewer, the oxigen atom of the the O-acetyl group has been marked in red in both the inversion pathways. The new Scheme 4 has been pasted to the manuscript for revision file and added to the schemes and figures folder.
Reviewer - Scheme 5. Please mark the O atom at the up wedge of (R)-6 in red color.
Authors response - As for the previous note, the Scheme 5 has been modified by marking in red color the newly formed O-acetyl group of compound (R)-6. The new Scheme 5 has been pasted to the manuscript for revision file and added to the schemes and figures folder.
Reviewer - Rows 245-246. Although the signals of the two Ac groups are overlapped, it is possible to distinguish the maxima of the singlet signals in the 1H NMR spectrum.
Authors response - At line 245 – 246, the description of the signals related to the two acetyl groups of compound 6 has been corrected by changing “195 – 1.90 (m, 6H, 2 Ac)” with “1.93 (s, 3H, Ac), 1.92 (s, 3H, Ac)”.